# Electrical Contact Performance of Cu Alloy under Vibration Condition and Acetal Glue Environment

**DOI:** 10.3390/ma15051881

**Published:** 2022-03-03

**Authors:** Zhongqing Cao, Yanqing Yu, Liping He, Yuchen Nie, Congyu Gong, Xiaohong Liu

**Affiliations:** 1School of Mechanical Engineering, Southwest Jiaotong University, Chengdu 610031, China; zqcao@swjtu.edu.cn (Z.C.); nieyuchen00@gmail.com (Y.N.); g1051215805@163.com (C.G.); 2School of Materials Science and Engineering, Southwest Jiaotong University, Chengdu 610031, China; heliping@swjtu.edu.cn

**Keywords:** electrical contact, sliding wear, acetal glue, external vibration

## Abstract

In view of the serious sliding electrical contact performance caused by external vibration and environmental contaminant, a study on the tribological characteristic and contact resistance of Cu alloy was conducted using a self-developed micro-load reciprocating electric contact device. Various glue concentrations (0%, 10%, 30%, and 50%) were prepared with anhydrous ethanol and deposited on the surface of a pure copper block via the deposition method. An external vibration source was installed on the sliding module to achieve vertical vibration. The results indicate that the final contact resistance and coefficient of friction (COF) in direct metal contact are about 0.01 Ω and 0.3, respectively. At this time, the wear volume is 2 to 3 orders of magnitude higher than the condition with glue residual. As glue concentration is above 10%, residual glue on the surface of Cu alloy hinders efficient contact between friction pairs, resulting in higher contact resistance. Glue exhibits lubrication, anti-wear, and insulation properties. External vibration causes friction pairs to briefly separate, leading to a lower glue removal capacity than that under non-vibration conditions. The contact resistance with glue addition under vibration conditions is higher than that under non-vibration conditions at 3 × 10^4^ cycles. The dominant oxide product is CuO, which has a limited effect on contact resistance.

## 1. Introduction

Electrical contact components are indispensable in machine systems to realize the efficient transmission of current and electrical signal between components [1]. Their structure is divided into sliding, fastener, and detachable. The sliding electrical contact structure is widely used the slip ring, sliding potentiometer, and brush [2], for the stable transmission of the current and electrical signal during sliding [3,4]. Cu alloy had satisfactory electrical conductivity and was widely used in sliding electrical contact structures [5]. However, acetal glue, commonly used for the packaging of electric contact components, will accelerate evaporation and stick to the surface of the exposed electrical contact component at higher service temperatures [6]. Volatile glue absorbed on the surface of electrical contact component is considered an environmental contaminant [7]. The stability of electrical signal transmission on the Cu alloy decreases dramatically under an acetal glue environment, leading to a decrease in the reliability of electrical components, especially for some milliamp precision aviation or military equipment.

A series of researches have been carried out on electrical contact with environmental contaminants. Frank et al. [8] studied the electric contact performance of various materials for electrical power connectors under a cyclic salt fog environment and found that all-copper connectors present a satisfactory electrical contact performance after 2000 h corrosion. Zhang et al. [9] investigated the influence of environmental contaminants on electrical contact performance during fretting wear. The formed environmental contaminant reduces the effective conductive area and increases contact resistance, eventually resulting in contact failure [10,11]. Ray et al. [12] revealed that contact resistance is independent of nominal contact. An environmental contaminant accumulates in the contact interface and hinders the direct contact of metals. The actual area of contact pairs remarkably contributes to the contact resistance.

Precision aviation or military equipment are also inevitably affected by external vibrations in actual operation [13]. External vibrations usually change electrical contact and degrade electrical contact performance [14]. When the amount of electrical contact reduces due to the momentary separation of electrical contact under vibration conditions [15], contact resistance increases and current sharply decreases [16]. When the separation gap is extremely large, arc erosion occurs and reduces the electrical contact stability [17,18]. Khomenko et al. [19] reported that external vibration and shock accelerate the degradation of battery internal components. Swingler et al. [20] built an external vibration platform and found that contact resistance is within the satisfactory level before the separation of electrical contact. The stability and reliability of the sliding electrical contact structure are seriously affected by environmental contaminants and external vibration.

Previous studies focused on electrical contact rather than the wear mechanism during sliding [21,22]. The mechanism of sliding electrical contact under external vibration conditions is not unclear, especially when an environmental contaminant is added. Therefore, this work studied the combined action of glue and external vibration on electrical contact performance. Analysis was conducted on the coefficient of friction (COF), contact resistance, wear scar morphology, and wear volume.

## 2. Materials and Methods

### 2.1. Sliding Electrical Contact Test Procedure

Sliding wear experiment was performed on a self-developed micro-load reciprocating electric contact device [23], as shown in Figure 1. The liner motor drove the pure copper block on the sliding module to achieve reciprocating sliding in the horizontal direction. An external vibration platform was mainly composed of cam and servo motor, which was installed on the sliding module to realize vertical vibration of pure copper block. The sliding electrical contact under external vibration condition was obtained between the interface of friction pairs. Another servo motor drove the loading device to move downward and fixed when normal load reached setting value. Normal load and friction force were collected by a force sensor.

The contact resistance measurement system was composed of friction pairs, current regulated inverter, voltage, and a constant resistance (*R*) of 10 Ω. Current regulated inverter outputted a constant current (*I_c_*). The voltage (*V*) was recorded by a voltmeter. Thick glue or large vibration might cause high contact resistance at the interface of friction pairs. Transient high voltage occurred at constant current when constant resistance was connected in series, resulting in damage to voltmeter. Therefore, constant resistance was measured using the improved four-wire method [24], in that constant resistance was connected in parallel to avoid transient high voltage, as shown in Figure 1b. The calculated contact resistance (*R_m_*) was governed by the following equation:(1)$$R_{m}=\frac{1}{\frac{I_{c}}{U}-\frac{1}{R}}$$

Two vibration amplitudes and four glue concentrations were selected to analyze their coupling effect on wear mechanism and electrical contact performance. Detailed parameters during sliding wear were shown in Table 1.

### 2.2. Material and Glue Design

Pure copper block (Cu, 99.9%; Sn, 0.002%; Zn, 0.005%; Pb, 0.005%; Fe, 0.005%; S, 0.005%; and balanced total impurities) was used as substrate against with brass alloy wire (Cu, 63.5%; Fe, 0.01%; Pb, 0.08%; P, 0.015%; Sn, 0.005%; Zn; REM; and balanced total impurities). The dimension of pure copper block was 10 mm × 10 mm × 30 mm. The diameter and bending diameter of brass alloy wire were 1 mm and 5 mm, respectively. Prior to the experiment, the pure copper block was grinded and polished to mirror for sliding electrical contact test. Various volume fractions of glue on the surface of pure copper block were prepared via deposition method [25] to explore the effect of acetal glue concentration during sliding contact. Acetal glue x98-11 was produced from the same batch to ensure uniformity. The prepared process was as follows. (I) Anhydrous ethanol and glue were mixed in ratios of 18:2, 14:6, and 10:10 (mL). (II) Anhydrous ethanol and glue ultrasonically oscillated for 10 min. (III) One side of pure copper block was immersed in the solution for 1 min. (IV) Pure copper block was air-dried until anhydrous ethanol was completely volatilized and glue was uniformly distributed on its surface. Finally, the samples with glue concentrations of 10%, 30%, and 50% were obtained.

Before the experiment, the surface morphology and chemical composition on the surface of pure copper block at various glue concentrations were presented in Figure 2. Extra agglomeration occurred on the surface of sample in high concentration of glue addition [26]. Apparent agglomeration was observed when the glue concentration reached 50%. EDS results in Figure 2e–f showed that only a small amount of Cu element could be detected on the sample surface when the glue concentration was up to 30%. This situation indicated that uniform and compact glue with a certain thickness was deposited on the sample surface.

### 2.3. Material Characterization

Wear scar morphology and element distribution were analyzed using an optical microscope (OM; VHX-6000, KEYENCE, Osaka, Japan) and a scanning electron microscope (SEM; JSM-7001F, JEOL, Tokyo, Japan). Wear scar profile and wear volume were measured by a white light interferometer (Bruker, Contour GT-K1, Billerica, MA, USA). Cu valence was investigated using an X-ray photoelectron spectroscope (XPS, Thermo Fisher, ESCALAB Xi+, Waltham, MA, USA) to analyze its oxide.

## 3. Results

### 3.1. Response of Contact Resistance and COF

Figure 3 shows the COF at various glue concentrations with and without vibration. When no glue is added, the evolution of COF with cycles under non-vibration and vibration conditions has two stages, including a quickly ascending and stable stage. The COF at the stable stage is about 0.3 for two vibration conditions. However, the COF under non-vibration conditions enters the stable stage after 1 × 10^4^ cycles, while the COF under vibration conditions enters the stable stage after 5 × 10^3^ cycles. As the glue concentration reaches 10%, the COF under non-vibration conditions shows a similar trend as no glue addition. However, COF at a 10% concentration remains about 0.1 before 1.5 × 10^4^ cycles and then reaches 0.3. The increase in the COF after 1.5 × 10^4^ cycles may be ascribed as the complete removal of glue and direct contact of metals. The COF under vibration conditions is always lower than 0.2 at a 10% glue concentration. This situation implies a limited glue removal capacity under vibration conditions. Residual glue on the sample surface still acts as lubrication.

When the glue concentration is further increased to 30% and 50%, the COF remains essentially constant throughout the cycle. At this time, the COF at 3 × 10^4^ cycles with and without vibration is lower than 0.2, which further demonstrates the residue of glue and its lubrication effect. The COF in the first 2 × 10^3^ cycles increases with the increase of the glue concentration from 10% to 50%, which may be due to the positive correlation between viscosity and concentration of glue. Vibration and glue concentration will affect the removal of glue, resulting in various contact statuses between friction pairs.

The calculated contact resistance throughout the cycle under non-vibration and vibration condition is shown in Figure 4. The contact resistance of 10^4^ Ω is considered to have reached insulation, which is up to the voltmeter range. Under non-vibration conditions, the contact pairs without glue complete the running-in process in 5 × 10^3^ cycles and then maintain a contact resistance of about 0.01 Ω until the end of cycle. The contact resistance also reaches 0.01 Ω after 10^4^ cycles as the glue concentration increases to 10%. Similar to the COF trend, the decrease in contact resistance is due to the direct contact of friction pairs. The contact resistance is insulated in the initial stage and then maintains a value above 1 Ω in the following cycles when the glue concentration is higher than 30%. This situation also indicates that glue cannot be completely removed by friction and wear at a high concentration. Residual glue has persistent effects on electrical contact performance throughout the cycle.

Under vibration conditions, the contact pairs without glue also complete the running-in process in 5 × 10^3^ cycles and maintain stable contact resistance similar to that under non-vibration conditions throughout the cycle. As the glue concentration increases to 10%, the contact resistance decreases rapidly to 0.1 Ω in the first 5 × 10^2^ cycles due to the removal of glue, and then slowly increases from 1 Ω to 10 Ω after 2 × 10^3^ cycles. Residual glue at the interface still reduces the effective conductive area [27,28], leading to a higher contact resistance than that without vibration. The contact resistance at 30% and 50% concentrations remains insulated in the initial stage with a value of approximately 100 Ω in the following stage. Glue acts in the roles of lubrication and insulation, resulting in the COF being closely related to contact resistance. Vibration during the sliding wear affects the removal of glue, which also has an influence on contact resistance. When glue is added, the contact resistance under the vibration condition is higher than that under the non-vibration condition.

Figure 5 shows the entire data point of contact resistance at various cycles under typical glue concentrations and vibration conditions. When the glue concentration is 10%, the contact resistance at the first cycle fluctuates between 0.01 Ω and 10 Ω under both vibration and non-vibration conditions, as shown in Figure 5a,b. The contact resistance is stable at 0.01 Ω under the non-vibration condition and 10 Ω under the vibration condition after 10^4^ cycles. When the glue concentration reaches 50%, the contact resistance at the first cycle is insulated with and without vibration. The proportion of data points in the insulated state decreases when the cycle increases. The fluctuation of contact resistance between insulation and non-insulation indicates that both glue and metal friction pairs are involved in sliding wear. Glue is repeatedly squeezed in and out, and the contact resistance is not insulated after 1.2 × 10^4^ cycles under the non-vibration condition. However, a portion of insulated resistance exists throughout the entire data point under the vibration condition. Higher contact with vibrations at 10% and 50% concentrations is ascribed to the low ability to remove glue under the vibration condition.

### 3.2. Analysis of Morphology and Component

Figure 6 shows the overall wear scar under various glue concentrations and vibration conditions. A large amount of wear debris covers the surface of the wear scar when no glue is added, as shown in Figure 6a,b. The dominant wear mechanism without glue is adhesive wear. The contact form changes into panel–panel, and the actual contact area increases due to the material removal of friction pair. Therefore, the contact resistance remains stable at 0.01 Ω until the end of cycles. Wear morphology also changes when glue is added. At 10% glue concentration, part of the wear debris accumulates on the surface of the wear scar under non-vibration conditions, whereas the surface of the wear scar is relatively smooth under vibration conditions, as shown in Figure 6c,d. The difference in wear morphology with and without vibration at a 10% glue concentration is related to the material removal capacity. Glue is completely removed under non-vibration conditions. Wear debris forms on the surface of the wear scar due to the direct contact of the friction pair, and this finding is similar to the results of COF and contact resistance in Figure 4 and Figure 5. When the glue concentration is further increased to 30% and 50%, the center of the wear scar is smooth and some wear debris accumulates at its edge, indicating the transition in the wear mechanism with an increasing glue concentration.

Figure 7 presents the worn morphology and chemical composition of the wear scar after 10^4^ cycles under non-vibration conditions. The dominant wear mechanism at 0% and 10% glue concentrations is adhesive wear because the hardness of the Cu sample is soft. Plastic deformation coordinates applied displacement, resulting in material transfer and accumulation in the wear scar. As a result, distinct material accumulation is found in the center of the wear scar at 0% and 10% glue concentrations, as shown in Figure 7a,b. The Zn element belonging to the upper specimen is detected in the wear scar of the pure copper block, further confirming the material transfer at the interface of friction pairs. Besides, some grooves exist at the 10% glue concentration in Figure 7b. The characteristic of the groove is distinct along the sliding direction without material accumulation when the glue is further increased. This situation proves the change in wear mechanism from adhesive wear to abrasive wear as the glue concentration is above 30% [29]. Residual glue acts as lubrication and hinders direct metal contact, leading to a formed wear scar being discharged from the wear scar rather than accumulating on the surface of the wear scar.

Figure 8 presents the worn morphology and chemical composition of the wear scar after 10^4^ cycles under vibration conditions. There is also obvious material accumulation and transfer at the 0% glue concentration, as shown in Figure 8a,f. No groove or Zn element are detected when the glue concentration is 10%, indicating that material removal under the vibration condition is slightly lower than that under the non-vibration condition. Glue remains on the surface of the wear scar, which reduces the actual contact area and hinders current conduction. As a result, the contact resistance under the vibration condition is higher than that under the non-vibration condition, as shown in Figure 4. As the glue concentration is further increased to 30% and 50%, invisible grooves can also demonstrate weaker material removal under vibration conditions [30].

Figure 9 presents the XPS results in the center of the wear scar to analyze the oxide of Cu at various operating conditions [31]. The Cu 2p spectrum can be decomposed into two components at binding energies of 932 and 934 eV, which are indicative of metallic Cu and CuO, respectively [32,33,34]. XPS results show that the formed oxide product is CuO under all conditions. This black oxide accumulates in the center of the wear scar due to adhesive wear, and is discharged from the wear scar due to abrasive wear, as displayed in Figure 7. The influence of CuO on contact resistance is limited, although it is a compound with poor electrical conductivity [35]. Contact resistance is mainly related to the contact status.

Figure 10 shows the 3D morphology of the wear scar under non-vibration and vibration conditions. Only an obvious wear scar is found at the 10% glue concentration with non-vibration or no glue addition, as shown in Figure 10. The difference in 3D morphology with and without vibration at a 10% glue concentration proves a lower glue removal capacity under vibration conditions, which is responsible for lower COF and higher contact resistance under vibration conditions. As the glue concentration is above 30%, the thicker glue layer cannot be completely removed, leading to the difference in 3D morphology with and without vibration being slight.

The cross-sectional profile of the wear scar is shown in Figure 11. When no glue is added, the wear scar presents a W-shape under non-vibration conditions and a U-shape under vibration conditions, as shown in Figure 11. External vibration causes a momentary separation of the contact pairs. Hence, the adhesive effect decreases and effective material removal occurs at the interface. The wear mechanism changes when glue is added. Owing to the removal of adhesive under non-vibration conditions at a 10% glue concentration, the profile of the wear scar also exhibits a U-shape. By contrast, the profile of the wear scar is flat under other conditions. Residual glue on the wear scar plays the role of lubrication and anti-wear, which leads to slight wear on the contact region and high, fluctuating contact resistance (Figure 5).

Figure 12 shows the wear volume at various vibration conditions. When no glue is added, part of the wear debris is discharged due to the separation of the contact pair under vibration conditions, resulting in a higher wear volume than that under non-vibration conditions. Owing to the direct contact of the metal, the wear volume under non-vibration conditions with a 10% glue concentration or without glue addition is higher than that under the other conditions. When glue is added, the wear volume under vibration conditions is lower than that under non-vibration conditions. The combination of external vibrations and glue reduces material removal between the friction pairs.

### 3.3. Analysis of Wear Mechanism

Figure 13 presents the combined action of glue and external vibration on contact resistance during sliding wear. Contact resistance data in the first 10 cycles at 2 × 10^4^ cycles without glue addition are shown in Figure 13a. The value of contact resistance is approximately 0.01 Ω with and without vibration when no glue is added. Part of the contact resistance reaches a large value under vibration conditions, which can be ascribed to the momentary separation of the friction pairs. The contact state without glue addition under no vibration and vibration conditions is shown in Figure 13b,c. The number of asperity in the contact state decreases because of the separation of the friction pair, eventually leading to a reduction in the actual contact area [36,37]. At this time, the current density and contact resistance increase under vibration conditions. The average contact resistance under vibration conditions is higher than that under non-vibration conditions. The momentary separation of the friction pair also reduces the wear between the friction pairs.

There are two contact statuses between the friction pairs including direct metal contact and glue residue. The difference in the wear mechanism with and without residual glue is shown in Figure 13d,e. Glue can be completely removed at 10% concentrations under no vibration condition. At this time, the contact state changes to direct metal contact, similar to no glue addition. The dominant wear mechanism is adhesive wear when the metal material directly contacts. Wear debris accumulates on the surface of the wear scar, accompanied by a material transfer, as shown in Figure 13d. The contact resistance decreases, and COF increases. Residual glue between the friction pairs plays the roles of lubrication, anti-wear, and insulation. At this time, the wear volume of the pure copper block is 2 to 3 orders of magnitude lower than that in direct metal contact, as shown in Figure 12. The brass alloy wire and pure copper block have a similar hardness, indicating obvious material removal of the brass alloy wire in direct metal contact. Part of the material from the brass alloy wire adheres on the surface of the pure copper block. Therefore, the Zn element is detected on the wear scar of the pure copper block in direct metal contact, as shown in Figure 7 and Figure 8.

According to the XPS results in Figure 9, the dominant oxide product is black CuO with a poor conductivity that covers the surface of the wear scar. However, the contact resistance reaches 0.01 Ω when no glue is added (Figure 4), implying the limited effect of CuO on contact resistance. Direct metal contact promotes current transmission. Most of the wear debris is discharged from the wear scar, and contact resistance is large at high glue concentrations. Therefore, residual glue is the main factor for high contact resistance.

## 4. Conclusions

The influence of glue and external vibrations on electrical contact performance was investigated using a self-developed micro-load reciprocating electric contact device. COF, contact resistance, wear morphology, and chemical composition were compared under various operating conditions. The following conclusions were drawn:
Glue acts as the effect of lubrication, insulation, and anti-wear. The wear volume under the condition with glue residue is 2 to 3 orders of magnitude lower than that under the other conditions.Vibration causes the separation of friction pairs and reduces their stability. The wear capacity decreases, and contact resistance fluctuation increases under the vibration condition. Contact resistance under the vibration condition is always lower than that under the non-vibration condition when glue is added.CuO is the dominant oxide product caused by the sliding electric contact. However, its influence on contact resistance is limited. Residual glue is the main factor for the high contact resistance.


## Figures and Tables

**Figure 1 materials-15-01881-f001:**
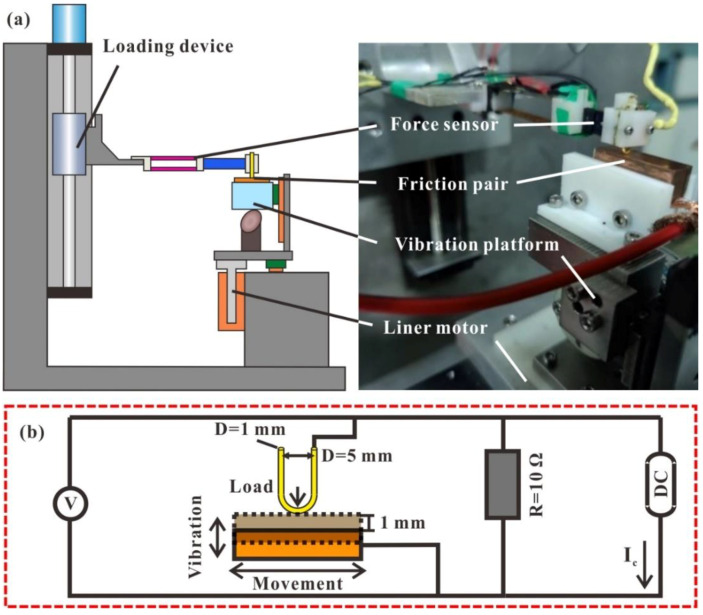
Schematic diagram of equipment: (**a**) Test device; (**b**) Contact resistance measurement system.

**Figure 2 materials-15-01881-f002:**
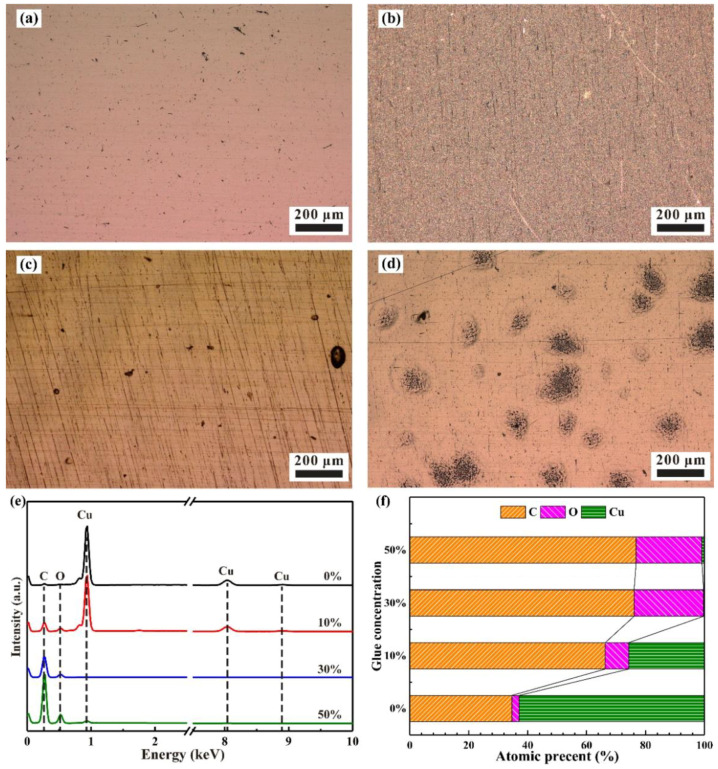
Surface morphology and chemical composition on the surface of pure copper block at various glue concentrations: (**a**–**d**) surface morphology at 0%, 10%, 30%, and 50% concentration, respectively; (**e**) chemical composition on the surface of pure copper block at various glue concentrations; (**f**) atomic percent corresponding to (**e**).

**Figure 3 materials-15-01881-f003:**
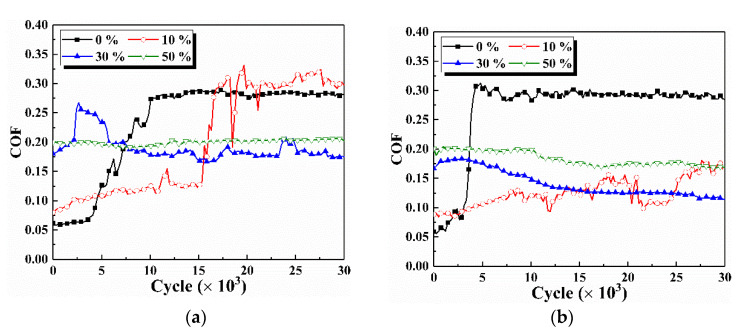
COF throughout the cycle under various vibration conditions. (**a**) Non-vibration, (**b**) Vibration.

**Figure 4 materials-15-01881-f004:**
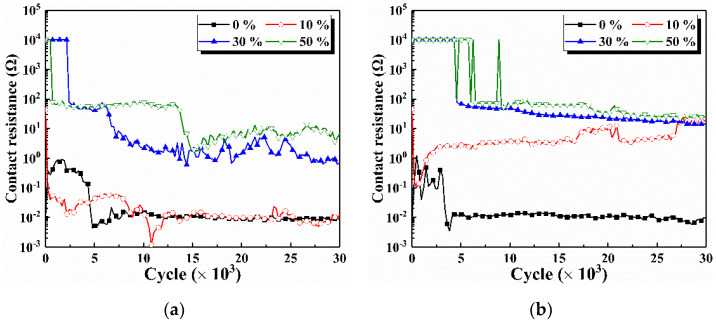
Contact resistance throughout the cycle under various vibration conditions. (**a**) Non-vibration, (**b**) Vibration.

**Figure 5 materials-15-01881-f005:**
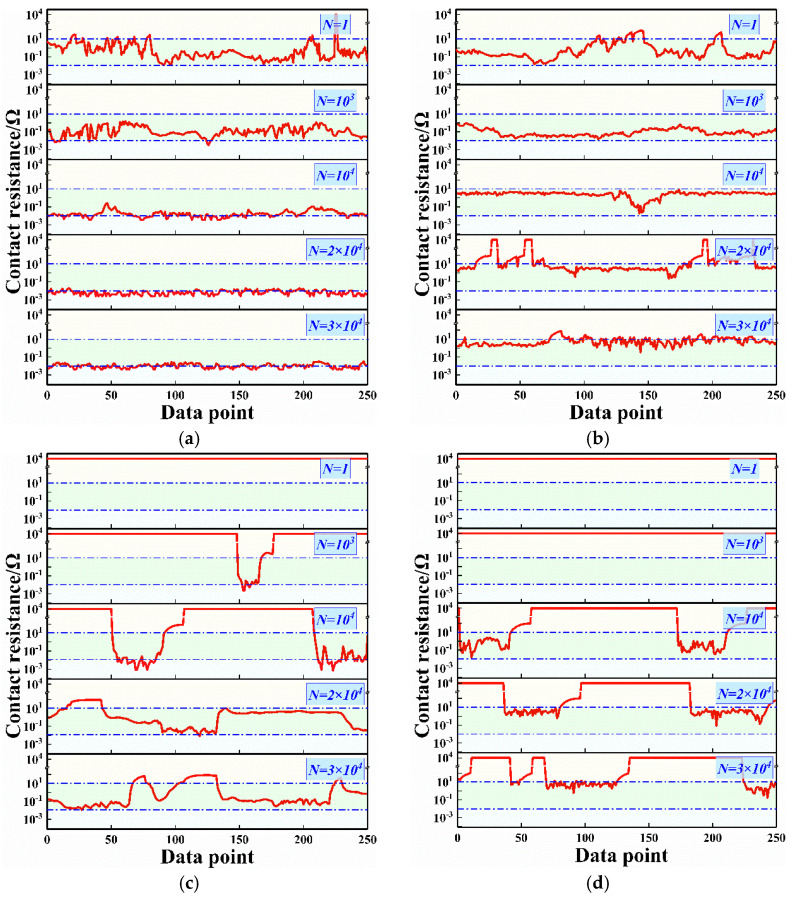
Entire data point of resistance at various cycles under typical glue concentrations and vibration conditions. (**a**) Non-vibration, 10% concentration, (**b**) Vibration, 10% concentration, (**c**) Non-vibration, 50% concentration, (**d**) Vibration, 50% concentration.

**Figure 6 materials-15-01881-f006:**
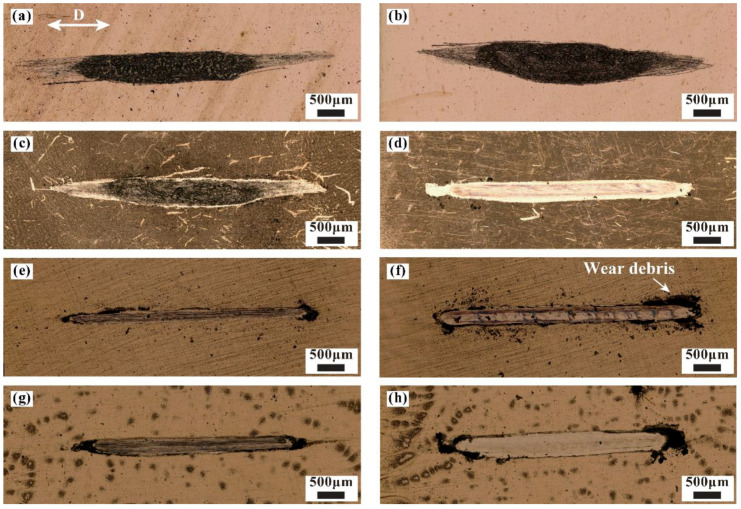
Overall wear scar under various operating conditions: (**a**,**c**,**e**,**g**) non-vibration and 0%, 10%, 30%, and 50% concentrations, respectively; (**b**,**d**,**f**,**h**) vibration and 0%, 10%, 30%, and 50% concentrations, respectively.

**Figure 7 materials-15-01881-f007:**
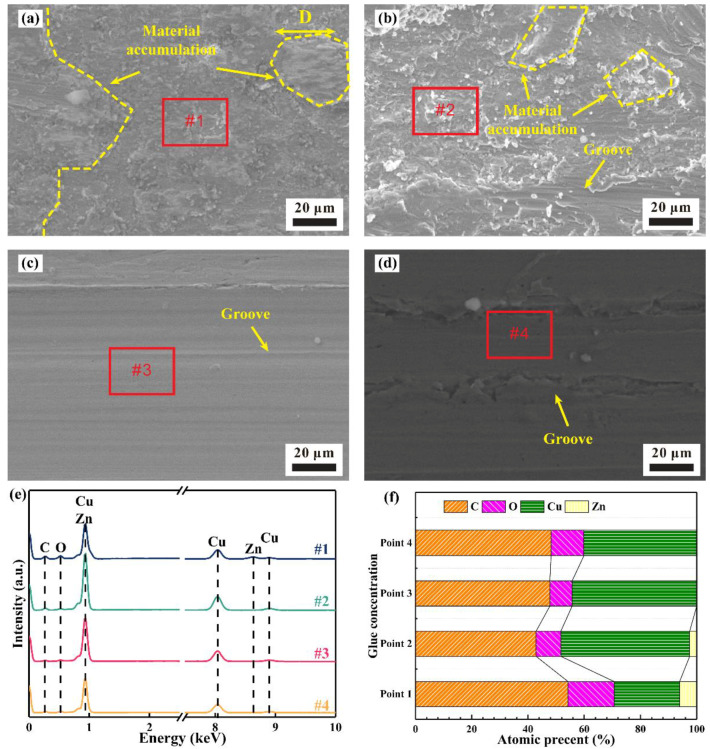
Worn morphology and chemical composition of wear scar under non-vibration condition: (**a**–**d**) wear scar morphology at 0%, 10%, 30%, and 50% concentrations, respectively; (**e**) chemical composition in the center of wear scar at various glue concentrations; (**f**) atomic percent corresponding to (**e**).

**Figure 8 materials-15-01881-f008:**
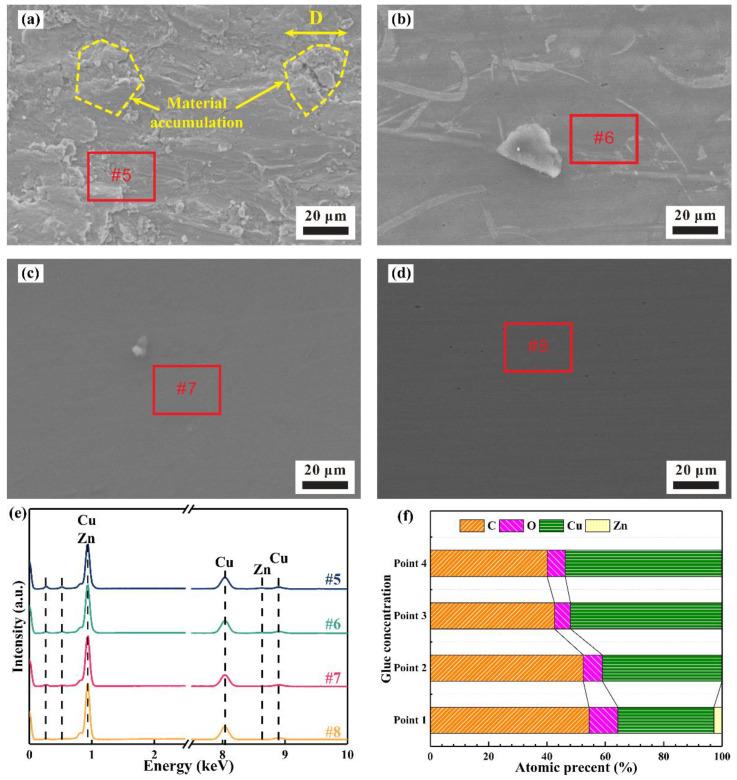
Worn morphology and chemical composition of wear scar under vibration condition: (**a**–**d**) wear scar morphology at 0%, 10%, 30%, and 50% concentrations, respectively; (**e**) chemical composition in the center of wear scar at various glue concentrations; (**f**) atomic percent corresponding to (**e**).

**Figure 9 materials-15-01881-f009:**
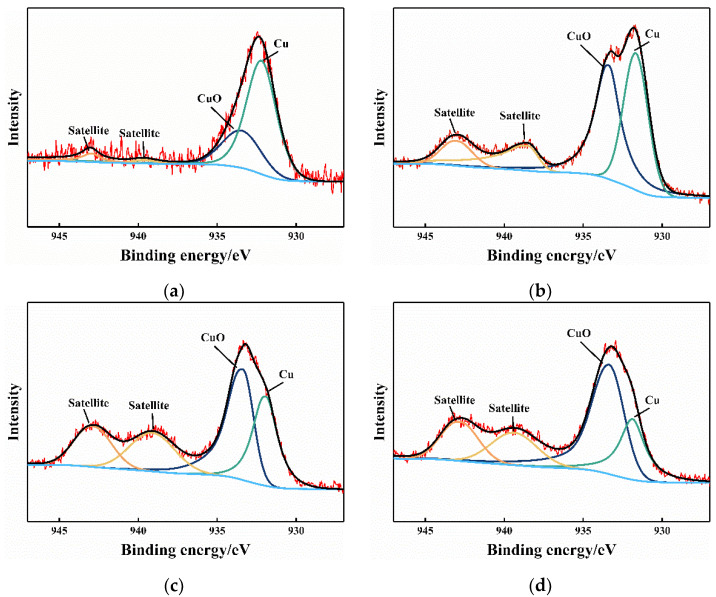
XPS high-resolution spectra of Cu 2p. (**a**) Non-vibration, 10% concentration, (**b**) Vibration, 10% concentration, (**c**) Non-vibration, 50% concentration, (**d**) Vibration, 50% concentration.

**Figure 10 materials-15-01881-f010:**
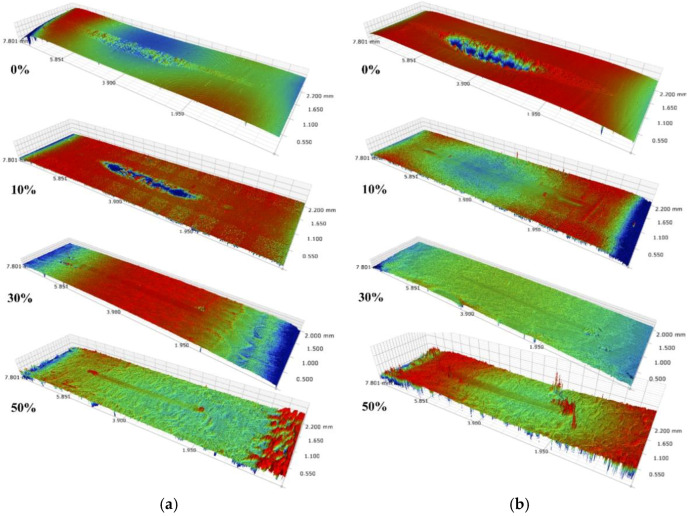
3D morphology of wear scar under various vibration conditions. (**a**) Non-vibration, (**b**) Vibration.

**Figure 11 materials-15-01881-f011:**
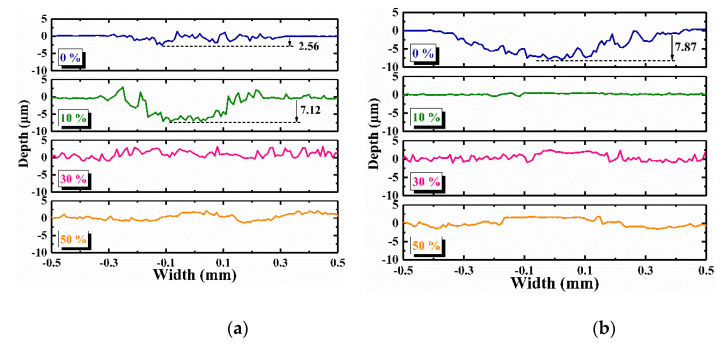
Cross-sectional profile of wear scar. (**a**) Non-vibration, (**b**) Vibration.

**Figure 12 materials-15-01881-f012:**
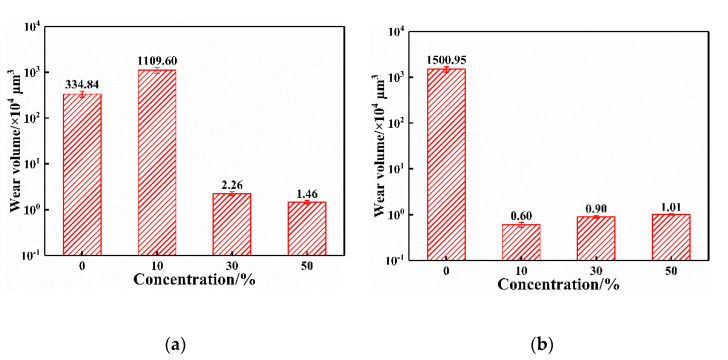
Wear volume under various conditions. (**a**) Non-vibration, (**b**) Vibration.

**Figure 13 materials-15-01881-f013:**
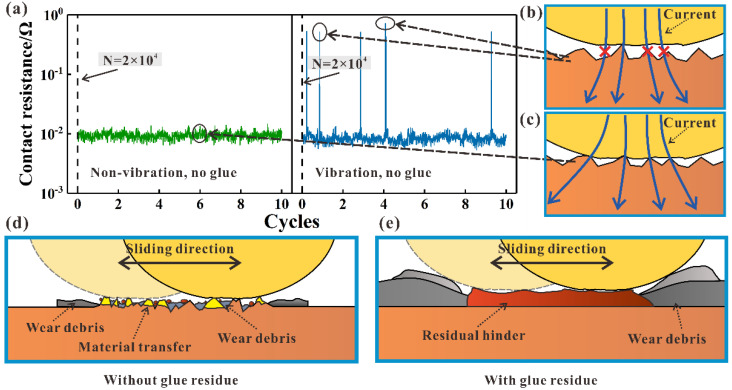
Combination of glue and external vibration on contact resistance during sliding wear: (**a**) collected contact resistance; (**b**–**e**) wear mechanism at various conditions.

**Table 1 materials-15-01881-t001:** Detailed parameters of the experiment.

Normal Load	Displacement Amplitude	Sliding Frequency	Current	Vibration Amplitude (Frequency)	Concentration of Glue
200 mN	6 mm	4 Hz	100 mA	0 mm (0 Hz)	0%
					10%
					30%
					50%
				1 mm (5 Hz)	0%
					10%
					30%
					50%

## Data Availability

The data presented in this study are available on request from the corresponding author.

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
