# Peer review of "Electrical Contact Performance of Cu Alloy under Vibration Condition and Acetal Glue Environment"

_materials, 2022, doi:10.3390/ma15051881_

Round 1

Reviewer 1 Report

Overall, the article is interesting, but it is not very clear what it is intended to demonstrate.  It would be necessary to better define what the "Acetal Aerosol Environment" is and what it is used for. Also specify in what kind of specific applications this study can be useful. Sliding electric mechanisms are very varied and can vary greatly between them, driving from milliamps to hundreds of amps in applications as different as consumer electronics or electric traction.  On the other hand, it is indicated that high temperatures favor the evaporation of aerosol [line 35], but in the development of the paper no reference is made to temperature again.

In my opinion, the design of the experiment, while seeming correct and useful, should be explained in more detail. Especially as far as the geometry of the copper contact is concerned. It is imperative to give a notion of the contact surface. In addition, the vibration values are not well quantified.

Another series of errors have been detected, which are discussed below:

[Lines] - Comment

[32] - Plenty of point after reference.

[34 - 35] - Reference to temperature.

[35] - Aerosol as a contaminant. What is the aerosol composed of? What is it used for?

[47] - ominal by nominal.

[59] – batter, I don’t understand.

[79] - On the improved method of measuring four-wire resistance. Explain in detail how the method is applied. Equipment used. Voltmeter and current generator. Models. Choice of values. Is 100 mA used in any case? Length of the measuring threads? Comparison of the measure with a pattern?

[81] - Utility of including a load resistor of 10 Ω in parallel.

[82] - High voltage transients occur by introducing a constant current at times when the contact resistance is very high. Why keep the current constant? Is it necessary to maintain the current of 100 mA during the mechanical test? Have the authors considered measuring the contact resistance after mechanical cycles?

[91] - On the values of charge and vibration. Why did the authors choose 200 mN of charge? What does a vibration frequency of 5 Hz at an amplitude of 0 mm mean? Vibration should be expressed in terms of acceleration.

[102] - Is it common in the industry to use anhydrous ethanol as an aerosol thinner?

[109] - What can be the origin of these impurities?

[118] - cu by Cu.

[131] - Include the two spaces in scientific notation. 1.5× 10^4 by 1.5 × 10^4.

[133] - I think authors should delve deeper into the COF. Definition, implications, trend...

[145] - Have you measured resistances of 10 kΩ with the method described?

[267] - Figure 12. The ordinate axis is not consistent with what is shown. It certainly gives a sample of the created profile, but it should include a zero and a measure of endpoints for each profile.

[276] - Figure 13. The measurement of the volume of displaced material is very interesting. How was it obtained? Is it the excess material at the edges of the groove or the defect of the groove itself? Is it possible that this material also comes from the sliding element? Can the material adhere to the sliding element? Again, the geometry of the contact is relevant. This is referred to in line 300, but it would be interesting to go a little deeper into this matter.

[276] - If they are not referenced, error bars should be removed.

[303] - Figure 14 - d. Wear dbris by Wear debris.

[303] - Why are current lines marked between contact and substrate? The reduction of contact surface will certainly increase the current density and with it the resistance. This may justify the dynamic measurement of resistance, but the drag and friction mechanisms are so brief that they should be studied carefully and at different loads and oscillations.

[308] - Does high-frequency refer to vibration or contact design?

[331] - Add space between China and (U1730131).

Reviewer 2 Report

Dear Authors,

Authors in this article describe research conducted so far in the subject of electrical contact performance of Cu alloy. Their research focused on the combined action of aerosol and external vibration on electrical contact performance. Analysis was conducted on coefficient of friction (COF), contact resistance, wear scar morphology, and wear volume.

The article contains a complete set of studies presented in a very neat way. All citations are correct. The drawings are correctly described and discussed in the text. The research presented in the article is interesting and well described. The only thing that can be considered a "minus" is that the reader must delve deep into the article and pay a lot of attention to find the essence. Too scarcely described research is a disadvantage, but if there is too much information, it is also not good. However, I do not consider this a disadvantage that disqualifies the article. Congratulations to the authors of the article and I accept the article in this form.

Kind regards,

Reviewer

Reviewer 3 Report

“Electrical Contact Performance of Cu Alloy under Vibration 2 Condition and Acetal Aerosol Environment” This is an interesting manuscript that fits well with the scope of the Journal. However, some items should be considered as follows:

  1. Typos and minor grammar errors are common in the manuscript. A considerable number of sentences are not comprehensible, some of which can be due to typos or grammar errors. About the manuscript, it is recommended to recheck the grammatical errors or typos.
  2. The abstract section should be containing specific obtained values of the data. “Various aerosol concentrations were prepared with anhydrous ethanol and deposited on the surface of pure copper block via deposition method.” Please specify the exact aerosol concentrations.
  3. I suggest removing the figure of the introduction section.
  4. Materials and methods section: “The surface morphology and chemical composition of pure copper block at various concentrations were presented in Figure 3.” The results and related figures should be replaced in the result and discussion section.
  5. The conclusions section should be replaced in the appropriate position of the results and discussion part. The conclusion should be more concise. This section is too summarizing. So I suggest providing more conclusions.

Round 2

Reviewer 1 Report

First of all, I want to acknowledge the effort and dedication of the authors in answering my questions. His explanations are clear and detailed. All my suggestions have been considered and included in the article. In addition, the doubts I had about the experiment have been resolved. Thanks. I believe that the paper in its current form can be published. Finally, I encourage the authors to continue working on this line and to present new papers with different contact configurations and other types of switches

Reviewer 3 Report

The manuscript is accepted in present state.